# Robotic versus Conventional Overground Gait Training in Subacute Stroke Survivors: A Multicenter Controlled Clinical Trial

**DOI:** 10.3390/jcm12020439

**Published:** 2023-01-05

**Authors:** Sanaz Pournajaf, Rocco Salvatore Calabrò, Antonino Naro, Michela Goffredo, Irene Aprile, Federica Tamburella, Serena Filoni, Andreas Waldner, Stefano Mazzoleni, Antonella Focacci, Francesco Ferraro, Donatella Bonaiuti, Marco Franceschini

**Affiliations:** 1Neurorehabilitation Research Laboratory, Department of Neurological and Rehabilitation Sciences, IRCCS San Raffaele Roma, 00163 Rome, Italy; 2IRCCS Centro Neurolesi Bonino-Pulejo, 98124 Messina, Italy; 3Stroke Unit, Policlinico Universitario G. Martino, 98123 Messina, Italy; 4IRCCS Fondazione Don Carlo Gnocchi, 50143 Florence, Italy; 5Laboratory of Robotic Neurorehabilitation (NeuroRobot Lab.), Spinal Rehabilitation Laboratory (SPIRE Lab.), Neurorehabilitation 1 Department, IRCCS Fondazione Santa Lucia, 00124 Rome, Italy; 6Fondazione Centri di Riabilitazione Padre Pio Onlus, 71013 San Giovani Rotondo, Italy; 7Department of Neurorehabilitation, Melittaklinik Hospital, 39100 Bolzano, Italy; 8Department of Electrical and Information Engineering, Politecnico di Bari, 70121 Bari, Italy; 9S.C. Medicina Fisica e Riabilitazione, ASL 4 Ospedale di Sestri Levante, 16124 Genova, Italy; 10Struttura Complessa di Riabilitazione Neuromotoria ASST Mantova Presidio di Bozzolo, 46012 Bozzolo, Italy; 11Italian Scientific Society of Physical Medicine and Rehabilitation (SIMFER), 00198 Rome, Italy; 12Department of Human Sciences and Promotion of the Quality of Life, San Raffaele University, 00123 Rome, Italy

**Keywords:** stroke, gait, rehabilitation, robotics, exoskeleton, end-effector

## Abstract

Background: Although stroke survivors can benefit from robotic gait rehabilitation, stationary robot-assisted gait training needs further investigation. In this paper, we investigated the efficacy of this approach (with an exoskeleton or an end-effector robot) in comparison to the conventional overground gait training in subacute stroke survivors. Methods: In a multicenter controlled clinical trial, 89 subacute stroke survivors conducted twenty sessions of robot-assisted gait training (Robotic Group) or overground gait training (Control Group) in addition to the standard daily therapy. The robotic training was performed with an exoskeleton (RobotEXO-group) or an end-effector (RobotEND-group). Clinical outcomes were assessed before (T0) and after (T1) the treatment. The walking speed during the 10-Meter Walk Test (10 MWT) was the primary outcome of this study, and secondary outcomes were the 6-Minute Walk Test (6 MWT), Timed Up and Go test (TUG), and the modified Barthel Index (mBI). Results: The main characteristics assessed in the Robotic and Control groups did not differ at baseline. A significant benefit was detected from the 10 MWT in the Robotic Group at the end of the study period (primary endpoint). A benefit was also observed from the following parameters: 6 MWT, TUG, and mBI. Moreover, patients belonging to the Robot Group outperformed the Control Group in gait speed, endurance, balance, and ADL. The RobotEND-group improved their walking speed more than the RobotEXO-group. Conclusion: The stationary robot-assisted training improved walking ability better than the conventional training in subacute stroke survivors. These results suggest that people with subacute stroke may benefit from Robot-Assisted training in potentiating gait speed and endurance. Our results also support that end-effector robots would be superior to exoskeleton robots for improving gait speed enhancement.

## 1. Introduction

Gait disorders represent a serious consequence of a stroke: more than 75% of those affected lose the ability to walk [1,2] owing to the deterioration of one or more of the main determinants of mobility, including endurance (namely, resistance to walking), gait speed, and balance, together with cognitive functions [3,4,5,6,7]. These impairments negatively affect the quality of life, reduce autonomy, and consequently limit participation in social life [8]. Thus, restoring walking ability through specific approaches of gait training is one of the essential goals in neurorehabilitation after stroke [9,10,11]. There is strong evidence for a rehabilitation approach based on intensive, repetitive, assisted-as-needed, and task-oriented training in all the phases of post-stroke rehabilitation [9]. On the other hand, rehabilitation sessions aimed at gait recovery are extremely demanding from a physical and cognitive point of view, often requiring more than one therapist per patient, leading to a heavy burden on resources. In this context, various methods and technologies have been explored over the years [12,13,14] to lighten the work of physical therapists, including body weight support devices, and to intensify the treatment by allowing highly repetitive, progressively intensive, and programmable task-specific exercises, depending on the patient’s needs [15,16].

Robotic-assisted training plays a critical role in providing solutions to improve mobility. The devices currently available for robotic training exploit the principles of motor learning (as a result of finalized motor activities provided with high intensity) that involve the patient’s active motor and cognitive participation [7,17]. They can be categorized into end-effector systems and exoskeletons based on the type of mechatronic design and human–machine interaction. In the end-effector robots, the contact between the mechanical structure and subject is limited to the effector (e.g., the pedal). Exoskeletons, on the other hand, are wearable systems that simulate the same type of human movement (e.g., the robot has the same joints with the same amplitudes). Moreover, the robots for gait training can be categorized into stationary and overground devices.

Several randomized controlled trials have been published on the use of robots in stroke survivors [18]. A recent study showed that people with stroke who receive robot-assisted gait training in combination with standard physical therapy achieve positive effects in terms of independent walking, as compared with those receiving only conventional gait rehabilitation training [19]. However, several studies have reported inconclusive results regarding the efficacy and potential indications of using robotics alone in both acute and chronic stroke [20,21,22,23,24,25,26,27]. Additionally, a variety of devices and treatment duration and frequency contribute to the wide variability in outcomes across studies [21]. Systematic reviews found that individuals in the first three months after stroke and those who cannot walk appear to benefit most from this type of intervention [22,23], but more evidence is needed. Furthermore, few trials have been undertaken to compare the efficacy of different systems. Goffredo et al. [18] studied the effects of robot-assisted gait training using an end-effector and an overground exoskeleton compared with conventional gait rehabilitation on a small sample of stroke patients (*n* = 26; stroke onset of <6 months). The authors found significant improvement in functional outcomes in favor of robot-assisted gait training groups (with no significant intergroup changes), but no significant variations were found in gait spatial and temporal parameters. However, the comparison was limited to devices offering different environmental stimuli with different control strategies, i.e., the G-EO System^TM^ (stationary end-effector; device-in-charge) and the Ekso (overground free exoskeleton; patient-in-charge). In addition, a systematic review [24] illustrated the effects of different robotic devices for gait training, categorized by two control strategies on walking ability and balance in persons with chronic stroke. Therefore, there is an increasing need in the literature to investigate the specific effects of rehabilitation treatments based on different types of robotic devices for gait rehabilitation in stroke survivors. This will allow for optimizing their use and indications in rehabilitation and defining guidelines for the implementation of standardized therapeutic rehabilitation protocols.

The current study was designed with the aim of assessing the clinical effects of robot-assisted gait training compared with conventional training in subacute stroke patients. Considering that gait speed is a key indicator of post-stroke gait performance and has considerable importance on disability burden reduction [25], we hypothesized that gait speed would improve most in subjects who received an effective number of robot-assisted gait training sessions (a total of 20 sessions) [26,27] versus those treated with overground gait training. As a secondary analysis of data from patients receiving robot-assisted gait training, we compared the effects of the two types of stationary robotic devices.

Therefore, the main objective of this study was to evaluate the effects in terms of gait speed (measured by the 10-Meter Walk Test) of robot-assisted gait training compared with overground gait training in subacute stroke patients. Furthermore, we investigated whether the effects induced by end-effector robots differ from those induced by the exoskeletons.

## 2. Materials and Methods

### 2.1. Study Design

This study was based on a multicenter, single-blind, prospective, controlled clinical trial conducted between June 2018 and October 2020 in eight intensive subacute Italian rehabilitation hospitals with staff and resources dedicated to robotic rehabilitation.

### 2.2. Participants

Individuals admitted to the participating centers for rehabilitation treatment after a first-ever stroke in the subacute phase (≤6 months since stroke onset) were observed. Adults (both men and women) were eligible for this study if they met the following inclusion criteria: age ≤ 85 years; first occurrence of the pyramidal syndrome; ability to understand and perform simple instructions; anthropometric characteristics satisfying the requirements for the use of the robots.

Participants with the following criteria were excluded from this study: age > 85 years; bilateral impairment; Walking Handicap Scale (WHS) of <5 prior to the stroke event, which was collected as anamnestic data (patients unable to walk independently prior to the stroke onset due to any pre-existing orthopedic or neurologic disease); cognitive or behavioral deficits that interfere with understanding robotic training; neurolytic treatment with botulinum toxin in the past 3 months and/or during this study; use of other lower limb rehabilitation technologies during this study; inability or unavailability to provide informed consent; cardiorespiratory severity morbidity.

Patients were randomly assigned (based on medical record number) to a Robotic or Control Group. The Robotic Group was submitted to robot-assisted gait training with either a fixed exoskeleton (RobotEXO-group) or an end-effector device (End-effector Group, RobotEND-group), depending on the facilities and resources available at the rehabilitation hospitals involved in this study. The Control Group received the same amount of conventional overground gait training. The uniformity of treatment and sessions was maintained with respect to a single protocol approved by the Local Ethical Committee and managed during this study’s period with the organization of periodic update meetings.

### 2.3. Rehabilitation Intervention

All individuals with stroke admitted to the involved rehabilitation hospitals and eligible for this study received a gait training protocol (robotic or conventional). All interventions were performed by experienced physical therapists. In all groups, gait training was combined with standard daily therapy (no gait training sessions), including physical therapy (e.g., upper limb rehabilitation, functional task practice, and muscle strengthening), speech therapy, and/or occupational therapy as part of the tailor-made multidisciplinary rehabilitation project.

#### 2.3.1. Stationary Robot-Assisted Gait Training

Both robotic systems used for this study are stationary devices that do not provide free overground walking [28] and are characterized by the ability to provide programmable body weight support as well as speed and stride length with a device-in-charge control strategy. Participants in the Robotic Group received stationary robot-assisted gait training using a commercial exoskeleton (Lokomat Pro, Hocoma AG, Volketswil, Switzerland) or an end-effector gait trainer (G-EO System^TM^, Reha Technologies, Genoa, Italy).

The robotic training consisted of twenty sessions (3–5 days per week) lasting 30 min each. Concerning the robotic training, we used the following parameters: initial speed of 0.9 km/h; an increase in speed up to a maximum of 2.5 km/h; weight support less than 40–45% of baseline body weight; gradual and progressive reduction of weight support as appropriate.

#### 2.3.2. Conventional Overground Gait Training

The conventional overground gait training consisted of twenty sessions (3–5 days per week) lasting 30 min each. It included any technique or approach aimed at achieving control over postural passages, such as from sitting to upright standing to lateral and anteroposterior load transfer in orthostasis and reorganization of the stride from assisted to protected walking with parallels, and then through various assistive devices (i.e., a walker, tripod cane, or crunches).

### 2.4. Outcome Measures

All participants were clinically assessed at baseline (T0) and at the end of the treatment (T1) by the same physicians who were not involved in any other phase of this study’s protocol.

The primary outcome of this study was the gait speed measured using the 10-Meter Walk Test (10 MWT) [29].

The secondary outcomes related to walking ability were: the Timed Up and Go Test (TUG) [30] as an indirect measure of walking speed and balance; and the 6-Minute Walk Test (6 MWT) [31] as a measure of walking endurance. Further, secondary measures were obtained using the following ordinal and non-ordinal scales: Motricity Index of the affected Lower Limb (MI-LL) [32] to measure segmental lower limb strength; Modified Ashworth Scale of the affected Lower Limb (MAS-LL) [33] for spasticity (plantar flexor, knee extensor, and knee flexor spasticity measurements); Functional Ambulation Category (FAC) [34,35] to evaluate how much support the patient requires when walking, regardless of whether or not they use a personal assistive device; modified Barthel Index (mBI) [36,37] to measure the performance in activities of daily living and indoor activities.

The Walking Handicap Scale (WHS) assesses the level of participation in both indoor and outdoor walking [38]. Initially, the WHS score was collected as anamnestic data to evaluate the patient’s eligibility for this study and then administered as an outcome measure at the end of treatment (T1) only.

### 2.5. Ethical Aspects

Reporting of this controlled clinical trial follows the CONSORT statement. This study’s protocol was approved by the Ethics Committee (protocol code RP 01/2018; date of approval: 8 November 2018) and a priori registered on ClinicalTrial.gov (number: NCT03688165). Each participant signed an informed consent form before any study-related procedures. Each record in the database was identified by a unique alphanumeric code to protect the participants’ privacy. This study was in accordance with the Declaration of Helsinki.

### 2.6. Statistical Analysis

The sample size was initially calculated considering a Minimal Clinically Important Difference (MCID) value of the 10 MWT (primary outcome) equal to 0.16 m/s, with an SD value of 0.2 m/s [39], and by assuming a two-tailed independent *t*-test with power equal to 0.8 and alpha equal to 0.05 [37]. A total of 87 participants were sufficient for statistical analysis when considering a drop-out rate of 10% and were equally distributed into three groups: 29 in the RobotEND-group, 29 in the RobotEXO-group, and 29 in the Control Group. We primarily considered three groups as we aimed to assess differences between the Robotic Group (subdivided into RobotEXO-group and RobotEND-group) and Control Group, consistent with the original trial description (number: NCT03688165). Subjects were therefore re-grouped into two starting from this assumption while considering the primary analysis of this study: the Robotic Group and the Control Group. As we are aware that this can negatively affect the statistical power of the analysis by creating a clear group imbalance (i.e., a nearly 2:1 ratio), this issue deserves mention in the limitation section. However, the unequal group distribution may have some advantages, including increasing patient acceptability of the trial and, therefore, recruitment rates, ethical reasons for maximizing participants’ exposure to the most advanced treatments, increasing the number of testable variables, gathering additional safety information, and preventively reducing the drop-out rate [38,39,40,41,42,43,44].

Data were analyzed using the Statview software 5.0. (Abacus Corporation, Baltimore, MD, USA), following an intention-to-treat analysis using the last forward method. For descriptive statistics, data were presented as frequency (with the relative percentage), mean with standard deviation (SD), and median with minimum and maximum values for the categorical, continuous, and ordinal variables, respectively. We used the Wilcoxon rank sum test (categorical data) or paired *t*-test (continuous data) to compare the demographics and baseline outcome measures. The Kolmogorov–Smirnov test was used to assess the normality of the distribution (all *p* > 0.05). For the main analysis (pre-post differences between the Robotic and Control groups’ gait speed measured by the 10 MWT), we conducted a one-way ANOVA. The level of statistical significance was set at *p* < 0.05 (two-tailed). Conditional on the F-value’s significance, post-hoc *t*-tests were used to locate significant differences between groups, applying Bonferroni’s correction for multiple comparisons. The same statistical tests were applied in the secondary analysis of data for investigating the potential differences between end-effector-based robotic training and exoskeleton-based training. The effect size was calculated using Cohen’s *d* coefficient (<0.2 small; 0.2–0.8 medium; >0.8 large) to infer the magnitude of changes.

### 2.7. Data Availability

Data associated with the paper are not publicly available but are available from the corresponding author upon reasonable request.

## 3. Results

A total of 168 subjects with subacute stroke admitted to rehabilitation centers were screened: 106 were eligible for this study and assigned to either the Robotic Group (RobotEND-group or RobotEXO-group) or Control Group. Eight subjects discontinued the treatment within the first 10 days of admission (i.e., before the start of treatment) because of medication changes or medical complications, whereas six subjects discontinued the treatment for reasons unrelated to the type of treatment for more than 3 sessions, and three subjects were transferred to other hospitals at their own request, as shown in Figure 1. Finally, data from 89 subjects were analyzed: 61 in the Robotic Group and 28 in the Control Group; the baseline data are summarized in Table 1. At baseline, both the Robotic and Control groups were statistically homogeneous in all demographic and clinical characteristics. Furthermore, the data of the two robotic groups (*n* = 30 in RobotEND-group; *n* = 31 in RobotEXO-group) at baseline are summarized in Table 2.

All participants in the Robotic Group tolerated the training, and no adverse events were reported.

Concerning the primary analysis (Table 3), statistically significant differences were found in the 10 MWT (primary endpoint), and in the following secondary endpoints: 6 MWT, TUG, and mBI. In detail, patients belonging to the Robotic Group outperformed the Control Group in gait speed, endurance, balance, and ADL performance.

The variation of 10 MWT values compared with baseline values in all individuals of the Robotic and Control groups is shown in Figure 2. Both groups obtained a clinically significant improvement in the 10 MWT (MCID > 0.16 m/s). Such an improvement was significantly greater in the Robotic Group than in the Control Group.

Following the secondary analysis, the subjects in the RobotEND-group achieved a greater improvement in 10 MWT values than those belonging to the RobotEXO-group (Table 4). Better performance of the RobotEND-group than the RobotEXO-group was found in gait endurance, balance, and ADL, but these outcomes did not reach statistical significance. The other outcome measures showed no significant difference. A good level of independence in community ambulation was observed in all subjects at T1. Higher WHS values were observed in the RobotEND-group (4 [3; 5]) than in RobotEXO-group (3 [2; 5]).

## 4. Discussion

This multicenter, controlled clinical trial was primarily carried out with the aim of assessing the clinical effects of stationary robot-assisted gait training compared with the conventional overground gait training in subacute stroke patients, considering gait speed as the primary outcome. As a secondary analysis, the effects of the two types of stationary robotic devices (exoskeleton or end-effector systems) have been analyzed. The Lokomat-Pro exoskeleton (Hocoma AG, Zurich, Switzerland) and the G-EO System^TM^ end-effector (Reha Technologies, Genoa, Italy) were the stationary robotic systems included in this clinical trial. These devices are among the most widely employed in clinical rehabilitation trials in our country and also the most frequently cited in the literature [24,31].

As we hypothesized, robot-assisted gait training provided subacute post-stroke patients with significant improvement in almost all the parameters compared to the conventional approach. Particularly, our data suggest the superiority of robotic training over conventional training in gait speed, as measured by the 10 MWT. Specifically, the improvement in 10 MWT at the end of the trial was greater than the MCID compared with baseline values and twice greater in the Robotic Group than in the Control Group. This outcome suggests the utility of stationary robot-assisted gait training in subacute stroke subjects. Furthermore, the impact of robotics was found to improve gait endurance (6 MWT) and overall disability burden (mBI) compared with overground gait training.

There are conflicting reports in the literature indicating that clinical outcomes after robotic training are often not superior to conventional therapy [41], although robotic training might be more effective than conventional training in improving specific post-stroke gait abnormalities [42]. Furthermore, it remains to be addressed whether the use of robotic devices in addition to conventional physical therapy compared to conventional physical therapy alone could be beneficial for post-stroke patients [43]. On the other hand, Taveggia et al. showed a significantly higher increase (above the minimal detectable change) in functional independence and gait speed (10 MWT) at the end of the treatment and at follow-up in patients receiving robot-assisted gait training compared with the overground gait training [27]. Other studies reported a smaller mean improvement (0.09 m/s) in gait speed compared with our data (0.30 m/s) [44]. Finally, a systematic review revealed that robotic training had a better or similar effect compared to conventional training in the post-stroke population. However, the meta-analysis of gait speed indicated no significant differences between robotics and conventional training, with the exception of chronic stroke survivors, who showed a slight positive effect of robotic training in their gait speed [45]. Overall, it seems that robot-assisted gait training, in addition to conventional therapy, is significantly better than stand-alone conventional overground gait training in subacute stroke patients in terms of gait speed, whereas such superiority is not appreciable when comparing stand-alone robotic training to stand-alone overground gait training [46]. Noteworthy, there is a consistent discrepancy between the available studies focusing on robot-assisted gait training versus overground gait training with respect to training programming and the robotic device employed. It is conceivable that the superiority of robotics could depend on the specific training features as well as on the characteristics of the sample. It has been observed that varying guidance force and body weight support assistance may affect mBI, whereas treadmill walking speed may affect TUG. Therefore, finely tuning robotic parameters may result in improvements with more functional outcomes than conventional training can achieve [47]. However, more research is necessary to assess the superiority of stand-alone robot-assisted gait training on stand-alone overground gait training since the available data in the literature only allow us to state the usefulness of robotics as an adjunctive therapy.

Our data are in agreement with the literature on the efficacy of stationary robot-assisted gait training in rehabilitating balance and gait endurance [27,48,49,50,51]. In particular, we found a significant increase of 6 MWT in the Robotic Group compared with the Control Group. In contrast, Hidler et al. found significantly greater improvement in walking speed and distance in subacute stroke participants who received conventional gait training compared with those trained with the Lokomat [49]. However, the 6 MWT is particularly challenging for stroke patients in the subacute phase, and its recovery takes a long time. In fact, during this phase, patients are easily fatigued when performing complex functional tasks, such as walking for long periods of time. Therefore, the conflicting results regarding the 6 MWT are not surprising. However, due to the small number of studies and their high heterogeneity, more research is needed to draw solid and more definitive conclusions.

Furthermore, TUG significantly improved gait speed in the Robotic Group compared with the Control Group at the end of treatment, demonstrating that intensive gait rehabilitation assisted by stationary robotic devices can provide improved functional mobility to subacute stroke patients.

We found no significant differences in lower extremity spasticity and muscle strength in participants treated with robotics compared to participants who received conventional training, although the Robotic Group had higher (but not statistically significant) levels of spasticity at baseline.

As a secondary analysis, we compared the effects of the two types of stationary robotic devices.

We found that the 10 MWT improved gait speed at the end of treatment more in the RobotEND-group than in the RobotEXO-group, thus suggesting a greater benefit of end-effectors in potentiating gait parameters in individuals with subacute stroke. This finding is in agreement with published data, while the effect of exoskeletons on gait performance is questionable in both the acute and subacute phases [50]. Similar conclusions were also found in another systematic review and meta-analysis comparing the effects of different robotic devices on the improvement of gait disorders after stroke [51]. The authors reported that end-effector devices produced a significant improvement in gait speed compared with a control group, but there was no evidence that exoskeletons were more effective than conventional therapy.

Nonetheless, no significant differences in terms of MAS-LL and MI-LL were found between RobotEND-group and RobotEXO-group, indicating that stationary robot-assisted gait training improved these body functions regardless of the type of device used. In contrast, Aprile et al. found a significant reduction in spasticity in the Robotic Group (using the same end-effector device) compared with conventional gait rehabilitation [40]. Furthermore, the Lokomat exoskeleton has been shown to reduce spasticity immediately after therapy, although this effect appears not to be maintained during subsequent follow-up evaluations [52].

Participants trained with the end-effector system achieved a better (but not statistically significant) TUG improvement than those treated with the exoskeleton, showing that the overall mobility (which requires both static and dynamic balance) can be enhanced regardless of the type of robotic device applied for gait training.

Finally, a good recovery of autonomy in community walking, assessed with WHS, was achieved in all participants. In particular, the RobotEND-group achieved the highest (although not statistically significant) values of the WHS scores.

Thus, our data indicate that both end-effectors and exoskeletons are equivalent with respect to gait improvement and speed. The specificity of this effect may be influenced by the complexity of post-stroke recovery mechanisms across different periods of post-stroke phases. The recovery process after a stroke is a combination of spontaneous and mediated processes: most spontaneous recovery occurs within the first 3 months after a stroke, but recovery patterns can vary depending on many complex factors. Indeed, the phenomena of neuroplasticity with functional recovery can be enhanced depending on the timing (in terms of precocity) and the type of rehabilitation treatment [9].

In our study, patients belonging to the RobotEND-group had a shorter distance from the acute event than those belonging to the RobotEXO-group. While this was not statistically significant, it may be clinically relevant. These data may confirm that weight-supported end-effector systems offer clinicians the earliest solution for gait rehabilitation compared with the other two approaches. This early treatment in the subacute phase of a stroke may have positively influenced recovery in our sample. According to a recent Cochrane systematic review, individuals unable to ambulate at baseline and those treated soon after stroke have the greatest benefits from the intervention in terms of gait performance [30]. On the other hand, despite a greater distance from the acute event, the RobotEXO-group had a lower baseline gait ability than the other two groups (10 MWT, 6 MWT, and FAC). Participants in this group with the same level of impairment (motor impairment was measured in all groups with MI-LL) had a lower functional ability. Therefore, the Lokomat may be more suitable for people with severe gait impairments.

In addition, it is necessary to emphasize the importance of considering the various parameters that can be set in robot-assisted gait training (for example, body weight support and guidance assistance) when assessing the effectiveness of treatments. This information can help clinicians develop a personalized robotic approach to gait rehabilitation. To date, there is no indication of specificity between the two devices used for robotic training. Specific studies are needed to determine which factors may contribute to the different effects of the two devices.

The main strength of our study is to have compared the effects of robotic versus conventional gait rehabilitation as well as two different robot-assisted devices used in the real-life neurorehabilitation environment. This approach allows the collection of relevant information for the design of future studies with an improved methodology (i.e., RCT). The use of ICF-based outcome measures is another important novelty of this study.

While promising, our study has some limitations to acknowledge; first, the relatively small sample size. We recruited two very small groups, including two different robotic approaches, which clearly diminished the statistical power of this study. However, this was due to the availability of the techniques in the centers involved in the trial. Furthermore, there was some heterogeneity between the samples in terms of clinical and demographic characteristics, specifically regarding a difference in the time since stroke onset and a variable degree of deficit at the individual level, including the 10 MWT. However, all these discrepancies were statistically insignificant. Another important limitation is the impossibility of being blinded and the potential bias of being trained with an innovative and more stimulating approach. The lack of follow-up represents another main limitation of this study; then, we are not able to estimate how long any benefit obtained from electromechanical gait training can last. However, we did not include a follow-up assessment in the design of this study because, after rehabilitation in Italy, most stroke survivors are admitted to a comprehensive rehabilitation facility where they are treated with a variety of multidisciplinary rehabilitation protocols, and it could bias the analysis on the impact of robot-assisted gait training at follow up.

## 5. Conclusions

Our study suggests that robot-assisted gait training may offer major advantages over conventional training in improving gait and endurance in subacute stroke patients. Specifically, end-effector robots appear to be more promising than exoskeletons in improving gait speed. Although we cannot exclude that these differences are also due to the phase of the disease and the severity of the functional impairment, we can state that people with subacute stroke who are unable to ambulate autonomously may significantly benefit from stationary robot-assisted gait training.

Large, controlled studies are still needed to compare the results between the two device types in more homogenous populations and to determine the optimal protocol design for maximum efficacy. Finally, our study also suggests that we need to better understand the conditions under which certain devices become widely used and effective in rehabilitation.

## Figures and Tables

**Figure 1 jcm-12-00439-f001:**
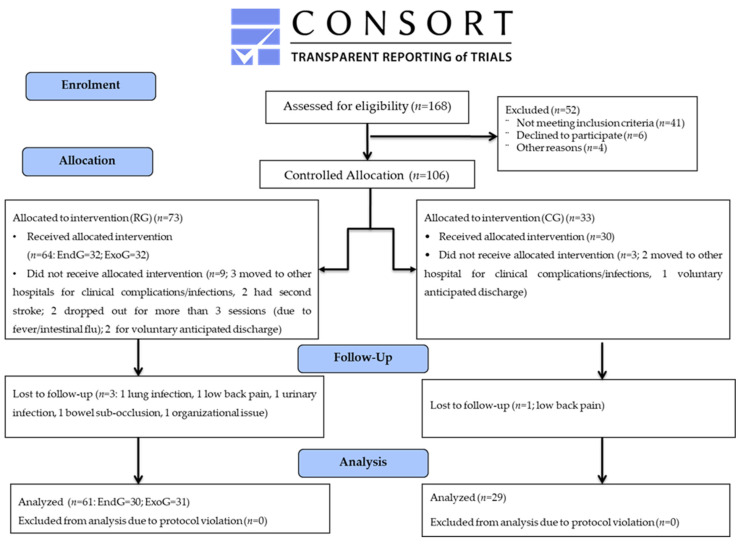
Consort flow-chart.

**Figure 2 jcm-12-00439-f002:**
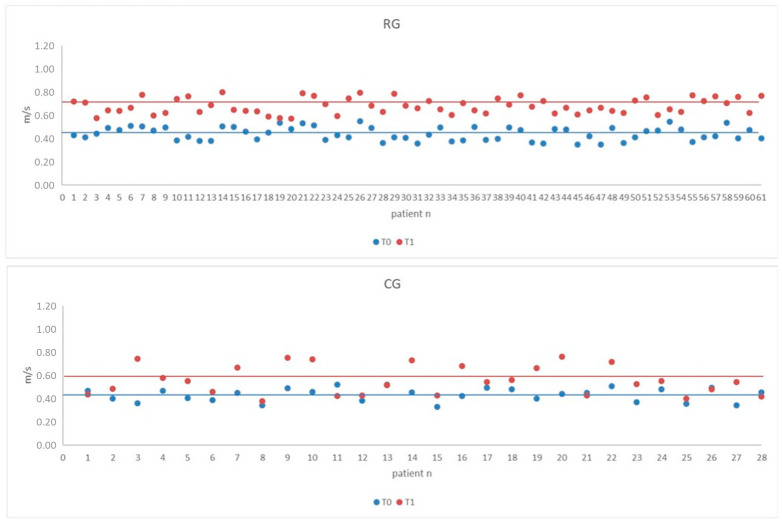
Scatterplot of 10-Meter Walk Test (10 MWT) of the Robotic Group (RG) and Control Group (CG) at baseline (T0) and final (T1) assessments. The horizontal lines within the plot represent the mean value.

**Table 1 jcm-12-00439-t001:** Demographic and clinical characteristics of the Robotic Group (RG) and Control Group (CG) at baseline (T0).

Variable	RG (*n* = 61)	CG (*n* = 28)	*p*-Value
Age (years)	59 ± 15	63.39 ± 12.85	0.2
Sex			0.9
Male	36 (59%)	18 (64%)	0.9
Female	35 (41%)	10 (36%)	0.9
Stroke onset (days)	48 ± 44	58.46 ± 43.24	0.2
Affected side			0.9
Right	28 (46%)	10 (35.7%)	0.1
Left	33 (54%)	18 (64.3%)	0.1
Etiology			0.2
Ischemic	53(87%)	26 (92.86%)	0.1
Hemorrhagic	8 (13%)	2 (7.14%)	0.2
10 MWT (m/s) *	0.43 ± 0.37	0.43 ± 0.46	0.4
MI-LL [0; 100]	45 [10; 80]	43 [1; 84]	0.2
MAS-LL [0; 12]	1.8 [0; 6]	0.5 [0; 5.5]	0.1
6 MWT (m)	102 ± 89	122 ± 158	0.9
TUG (s)	23 ± 14	23 ± 16	0.9
FAC [0; 5]	1.5 [0; 4.5]	2 [0; 5]	0.9
mBI [0; 100]	47 [11; 80]	43 [7; 91]	0.5

Abbreviations: Number (n); Robotic Group (RG); Control Group (CG); 10-Meter Walk Test (10 MWT); 6-Minute Walk Test (6 MWT); Motricity Index of Lower Limb (MI-LL); Time Up and Go Test (TUG); Modified Ashworth Scale of Lower Limb (MAS-LL); Functional Ambulation Classification (FAC); Modified Barthel Index (mBI); T0: Baseline assessment; T1: Final assessment. Notes: data are reported as N (%), mean ± SD, median [MIN; MAX], and * indicates primary outcome.

**Table 2 jcm-12-00439-t002:** Demographic and clinical characteristics of the End-effector Group (RobotEND-group) and Exoskeleton Group (RobotEXO-group) at baseline (T0).

Variable	RobotEND-Group (*n* = 30)	RobotEXO-Group (*n* = 31)	*p*-Value
Age (years)	58.03 ± 14.34	59.42 ± 15.51	0.2
Sex			0.9
Male	18 (60%)	18 (58.06%)	0.9
Female	12 (40%)	23 (41.94%)	0.8
Stroke onset (days)	33.30 ± 24.79	63.68 ± 49.92	0.2
Affected side			0.9
Right	17 (56.67%)	11 (35.48%)	0.1
Left	13 (43.33%)	20 (64.52%)	0.1
Etiology			0.2
Ischemic	27 (90%)	26 (83.87%)	0.1
Hemorrhagic	3 (10%)	5 (16.13%)	0.2
10 MWT (m/s) *	0.53 ± 0.38	0.33 ± 0.37	0.1
MI-LL [0; 100]	48 [1; 76]	43 [19; 84]	0.4
MAS-LL [0; 12]	2 [0; 6]	1.5 [0; 5]	0.2
6 MWT (m)	121 ± 88	83 ± 90	0.8
TUG (s)	23.38 ± 12.13	22.21 ± 16.20	0.9
FAC [0; 5]	2 [0; 5]	1 [0; 4]	0.8
mBI [0; 100]	48.5 [11; 75]	45 [12; 81]	0.6

Abbreviations: Number (n); End-effector Group (RobotEND-group); Exoskeleton Group (RobotEXO-group); 10-Meter Walk Test (10 MWT); 6-Minute Walk Test (6 MWT); Motricity Index of Lower Limb (MI-LL); Time Up and Go Test (TUG); Modified Ashworth Scale of Lower Limb (MAS-LL); Functional Ambulation Classification (FAC); Modified Barthel Index (mBI); T0: baseline assessment; T1: final assessment. Notes: data are reported as N (%), mean ± SD, median [MIN; MAX], and * indicates primary outcome.

**Table 3 jcm-12-00439-t003:** Clinical outcomes of the Robotic Group (RG) and Control Group (CG) at baseline (T0) and final (T1) assessments. Changes (Δ = T1–T0) in functional outcomes and the corresponding one-way ANOVA values are depicted.

Variable	Group	T0	T1	Δ	One-Way ANOVA *p*-Value (F)
10 MWT (m/s) *	CG (n = 28)	0.43 ± 0.46	0.57 ± 0.55	0.15 ± 0.26	**<0.0001** **(F = 17)**
	RG (n = 61)	0.43 ± 0.37	0.72 ± 0.44	0.30 ± 0.05	
MI-LL [0; 100]	CG (n = 28)	43 [1; 84]	76 [1; 100]	33 [27; 42]	**0.03** **(F = 4.87)**
	RG (n = 61)	45 [10; 80]	83 [38.5; 97]	38 [15; 50]	
MAS-LL [0; 12]	CG (n = 28)	0.5 [0; 5.5]	0 [0; 4.5]	−0.5 [0.25; 2.6]	0.7
	RG (n = 61)	1.8 [0; 5.5]	1 [0; 3]	−0.8 [0; −1.5]	
6 MWT (m)	CG (n = 28)	122 ± 156	172 ± 163	49 ± 26	**<0.0001** **(F = 112)**
	RG (n = 61)	102 ± 89	194 ± 110	92 ± 13	
TUG (s)	CG (n = 28)	23 ± 15.9	16.2 ± 7.9	−6.7 ± 1.5	**<0.0001** **(F = 109)**
	RG (n = 61)	22.8 ± 14.2	10.2 ± 5.2	−12.6 ± 4.81	
FAC [0; 5]	CG (n = 28)	2 [0; 5]	3 [0; 5]	1.0 [0; 4]	0.9
	RG (n = 61)	1.5 [0; 4.5]	3.5 [1; 6]	2 [1; 3]	
mBI [0; 100]	CG (n = 28)	43 [7; 91]	84 [20; 100]	41 [9; 13]	**<0.0001** **(F = 22.6)**
	RG (n = 61)	47 [11.5; 79.5]	89 [40; 95]	42 [25; 52]	
WHS [1; 6]	CG (n = 28)	n.a.	3 [1.5; 4.5]	n.a.	n.a.
	RG (n = 61)	n.a.	4 [3; 5]	n.a.	

Abbreviations: Number (n); Robotic Group (RG); Control Group (CG); 10-Meter Walk Test (10 MWT); 6-Minute Walk Test (6 MWT); Motricity Index of Lower Limb (MI-LL); Time Up and Go Test (TUG); Modified Ashworth Scale of Lower Limb (MAS-LL); Functional Ambulation Classification (FAC); Modified Barthel Index (mBI); Walking Handicap Scale (WHS). T0: baseline assessment; T1: final assessment; Δ = T1–T0. Notes: Significant *p*-values are reported in bold; data are reported as N (%), mean ± SD, median [MIN; MAX], and * indicates primary outcome.

**Table 4 jcm-12-00439-t004:** Secondary analysis: clinical outcomes of the End-effector Group (RobotEND-group) and Exoskeleton Group (RobotEXO-group) are reported as changes (Δ = T1–T0) of functional outcomes and the corresponding one-way ANOVA values.

Variable	RobotEND-Group (n = 30)	RobotEXO-Group (n = 31)	One-Way ANOVA *p*-Value (F)
Δ10 MWT (m/s) *	0.4 ± 0.4	0.2 ± 0.3	**0.0007 (F = 13)**
ΔMI-LL [0; 100]	14 [0; 76]	16 [0; 51]	>0.1
ΔMAS-LL [0; 12]	−1 [−3; 0]	−0.5 [−3; 1.5]	>0.1
Δ6 MWT (m)	112 ± 11	71 ± 89	>0.1
ΔTUG (s)	−6.84 ± 12.11	−4.55 ± 10.66	0.08
ΔFAC [0; 5]	1 [0; 4]	1 [0; 3]	>0.1
ΔmBI [0; 100]	29 [0; 70]	23 [−12; 65]	>0.1

Abbreviations: Number (n); End-effector Group (RobotEND-group); Exoskeleton Group (RobotEXO-group); 10-Meter Walk Test (10 MWT); 6-Minute Walk Test (6 MWT); Motricity Index of Lower Limb (MI-LL); Time Up and Go Test (TUG); Modified Ashworth Scale of Lower Limb (MAS-LL); Functional Ambulation Classification (FAC); Modified Barthel Index (mBI); Δ = T1–T0. Notes: Significant *p*-values are reported in bold; data are reported as N (%), mean ± SD, median [MIN; MAX], and * indicates primary outcome.

## Data Availability

The data associated with this paper are not publicly available but are available from the corresponding author upon reasonable request.

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
