# Peer review of "Robotic versus Conventional Overground Gait Training in Subacute Stroke Survivors: A Multicenter Controlled Clinical Trial"

_jcm, 2023, doi:10.3390/jcm12020439_

Round 1

Reviewer 1 Report (Previous Reviewer 3)

The authors have replied to the comments in an acceptable manner 

Author Response

Dear Reviewer,

Thanks for your comments.

We reviewed the manuscripts as you suggested.

Reviewer 2 Report (Previous Reviewer 2)

I reviewed author s’ response to my comments and I found that they considered all the comments in the new version which is perfect now and eligible for publication.

Author Response

Dear Reviewer,

Thanks for your comments.

We reviewed the manuscripts as you suggested.

This manuscript is a resubmission of an earlier submission. The following is a list of the peer review reports and author responses from that submission.

Round 1

Reviewer 1 Report

REVIEW for “Robotic rehabilitation versus traditional overground gait training in subacute stroke survivors”.

 General comments

The authors sought to compare different robotic-based rehabilitation modules with traditional overground gait training as part of the overall rehabilitation of subacute stroke survivors. Even though, the experimental design and methodology is sound, the authors should address the following points that would improve the overall quality of this study.

1.     Authors are advised to seek help from a strong (better native) English writer who is familiar with this field of work. This way they can improve the clarity of their paper, especially the introduction and part of the discussion sections. Methodology and Results are well written and clear.

2.     Authors have to rework on their statistical analysis. Why do you use ANCOVA for intragroup analysis? Intra- stands for within the groups analysis (pre-post design). I assume your intention was to use a 2X3 mixed two-way ANOVA model (time x group; pre-post and 3 different groups) or since you eliminated the pre-post design my calculating the difference, you can simply run an one-way ANOVA (3 groups). Performing t-test for time and t-test for difference among groups violates several assumptions and is inappropriate.

Author Response

Review Report #1

Comments and Suggestions for Authors:

The authors sought to compare different robotic-based rehabilitation modules with traditional overground gait training as part of the overall rehabilitation of subacute stroke survivors. Even though, the experimental design and methodology is sound, the authors should address the following points that would improve the overall quality of this study.

Thank you for your comments. We revised the manuscript following your suggestions.

  1. Authors are advised to seek help from a strong (better native) English writer who is familiar with this field of work. This way they can improve the clarity of their paper, especially the introduction and part of the discussion sections. Methodology and Results are well written and clear.

Thank you very much for this note. Based on your suggestions, we have revised the introduction and discussion to improve clarity.

  1. Authors have to rework on their statistical analysis. Why do you use ANCOVA for intragroup analysis? Intra- stands for within the groups analysis (pre-post design). I assume your intention was to use a 2X3 mixed two-way ANOVA model (time x group; pre-post and 3 different groups) or since you eliminated the pre-post design by calculating the difference, you can simply run an one-way ANOVA (3 groups). Performing t-test for time and t-test for difference among groups violates several assumptions and is inappropriate.

We thank the reviewer for this suggestion, which was adopted. Statistical analysis and result reporting were revised accordingly.

Reviewer 2 Report

The article entitled “Robotic rehabilitation versus traditional overground gait training in subacute stroke survivors” investigated the effect of robot-assisted gait training on functional recovery represented by lower limb spasticity and walking ability in subacute stroke survivors. The study compared the effect of two robotic devices namely the end-effector and the exoskeleton relative to the conventional rehabilitation therapy and found that the robotic devices effectively improved lower limb spasticity and walking ability. The study is very-well designed and organized and the outcome measures have been chosen precisely. However, there are several points that need to be addressed:

1.    In table 3, the authors stated that the outcome values have been obtained by subtracting the baseline (T0) values from the final assessment (T1) ones. However, the values of D MI-LL, D FAC, D mBI looked incorrect and need to be checked.

2.    Also, in table 3, it seems that the authors used ANOVA to compare between RG and CG, I don’t know if the different sample sizes were considered (RG group N=61; CG group N= 28).

3.    One of the exclusion criteria was Walking Handicap Scale (WHS) < 5 before the acute event. What do you mean by the acute event?

4.    Why was WHS not evaluated at T0 Like other outcome measure criteria? It was mentioned in the article that the value of WHS guided you to exclude some patients and that mean you have the measurements of WHS before training (at T0).   

Author Response

Review Report #2

Comments and Suggestions for Authors:

The article entitled “Robotic rehabilitation versus traditional overground gait training in subacute stroke survivors” investigated the effect of robot-assisted gait training on functional recovery represented by lower limb spasticity and walking ability in subacute stroke survivors. The study compared the effect of two robotic devices namely the end-effector and the exoskeleton relative to the conventional rehabilitation therapy and found that the robotic devices effectively improved lower limb spasticity and walking ability. The study is very-well designed and organized and the outcome measures have been chosen precisely. However, there are several points that need to be addressed:

Thank you for your comments. We revised the manuscript following your suggestions.

  1. In table 3, the authors stated that the outcome values have been obtained by subtracting the baseline (T0) values from the final assessment (T1) ones. However, the values of D MI-LL, D FAC, D mBI looked incorrect and need to be checked.

We apologise for this misreporting. Data and analysis were rebuilt as suggested by the reviewer 1. Data reporting was accurately checked. Please find the new results (table 2) highlighted in yellow.

  1. Also, in table 3, it seems that the authors used ANOVA to compare between RG and CG, I don’t know if the different sample sizes were considered (RG group N=61; CG group N= 28).

Thank you for your comment. As required also by the reviewer1, we rebuilt data analysis by running a one-way ANOVA with 3 groups and eliminating the pre-post design by calculating the difference, rather than a 2X3 mixed two-way ANOVA model (time x group; pre-post and 3 different groups). Please, find the new table 2 in the manuscript.

  1. One of the exclusion criteria was Walking Handicap Scale (WHS) < 5 before the acute event. What do you mean by the acute event?

We meant that patients unable to walk independently prior the stroke onset, due to any pre-existing orthopaedic or neurologic disease were not eligible for this study. We have added this detail to the methods.

  1. Why was WHS not evaluated at T0 Like other outcome measure criteria? It was mentioned in the article that the value of WHS guided you to exclude some patients and that mean you have the measurements of WHS before training (at T0).

The WHS-based exclusion criteria refer to the patient's independent mobility about the home and community prior to the stroke event. All participants had WHS > 5 before the stroke event, which was considered exclusively as anamnestic data. This detail has been clarified also in the “Methods” (subsections: participants and outcome measures).

The WHS evaluates the “participation” with respect to “International Classification of Functioning, Disability and Health (ICF)”, and includes ecological tasks such as going to a dentist appointment, visiting friends, shopping, going to church or to a restaurant, and travelling. Therefore, we believe the ideal setting for administering it during the hospitalization is when patients can return to frequent public areas such as the hospital’s bar, restaurant and park. In contrast, it could be inappropriate during the early phases of hospitalization.

Reviewer 3 Report

The present study which was to find the effect of robotic rehabilitation subacute stroke survivors. The manuscript provides a thorough, well designed report without however, much originality.

The authors have mentioned in the manuscript various studies already performed in this area. Moreover, robotic rehabilitation has been mentioned in various stroke rehabilitation guidelines. This questions the novelty of the general concept of the study. However, in this manuscript the authors were more specific towards comparing end effector robotic gait trainer and exoskeletal robotic gait trainer.

The manuscript is well documented in all the sections. As mentioned before it needs more justification for comparing robotic gait trainer with over ground gait training. Orelse the objective should be more specific towards comparison between exoskeletal gait trainer and end effector gait trainer. 

Needs a clarification whether all the centres had the facility to provide the robotic gait training. Needs more explanation of how this was managed, how the uniformity in treatment and sessions were maintained.

Page number 5 of 14 line number 219; I think it is mistaken as table 3.

Page number 2of 14; table 3: There was a significant difference in 10MWT between EndG vs CG and between EndG vs ExoG. However in 6MWT there was no significant difference in between group analysis among any of the groups. Authors could explain the probable reason for this even though both the outcome measures assess similar domains.

In the same table in abbreviations for Robotic Group it should RG instead of RT.

Author Response

Dear reviewer,

Thank you for your comments. We revised the manuscript following your suggestions. Attached, please find our point-to-point response to your comments.
